# Comparison of Mean Properties of Turbulent Pipe and Channel Flows at Low-to-Moderate Reynolds Numbers

Carmine Di Nucci [1,*] and Rafik Absi [2]

1   Civil, Construction-Architectural and Environmental Engineering Department (DICEAA), University of L'Aquila, Piazzale Ernesto Pontieri 1, Monteluco di Roio, 67100 L'Aquila, Italy

2   ECAM-EPMI–Ecole Supérieure d'Ingénieurs en Génie Electrique, Productique et Management Industriel, LR2E-Lab, Laboratoire Quartz (EA 7393), 13 Boulevard de l'Hautil, 95092 Cergy-Pontoise, France

*   Correspondence: carmine.dinucci@univaq.it

**Abstract:** We focus on the fully developed turbulent flow in circular pipes and channels. We provide a comparison of the mean velocity profiles, and we compute the values of the global indicators, such as the skin friction, the mean velocity, the centerline velocity, the displacement thickness, and the momentum thickness. The comparison is done at low-to-moderate Reynolds numbers. For channel flow, we deduced the mean velocity profiles using an indirect turbulent model; for pipe flow, we extracted the needed information from a direct numerical simulation database available in the open literature. A one-to-one comparison of these values at identical Reynolds numbers provides a deep insight into the difference between pipe and channel flows. This line of reasoning allows us to highlight some deviations among the mean velocity profiles extracted from different pipe databases.

**Keywords:** turbulence; internal flow; mean velocity profile; skin friction; indirect turbulence model

## 1. Introduction

In this note, we focus on the fully developed turbulent flow of Newtonian fluids in circular pipes and very large channels, both with smooth walls, at low-to-moderate Reynolds numbers. We do not consider the compressible effects [1–9]. We provide a comparison of the Mean Velocity Profiles (MVP), and we compute the values of the global indicators, such as the skin friction, the displacement thickness, the momentum thickness, the centerline velocity, and the mean velocity. The similarities and differences in turbulent pipe and 2D channel flows (referred to as internal turbulent flows) have generated considerable research attention. Previous results show that close to the wall, in the inner layer (the viscous sublayer plus the buffer layer, see, e.g., [10]), the MVPs are essentially indistinguishable [11–14]; in the outer layer, (the log-law layer plus the wake layer [10]) the MVPs show remarkable differences [15–17].

Coles [18] provides a comprehensive study on the MVPs in the outer layer. This classical paper is about investigating and modeling deviation of data from logarithmic law. The log-wake law proposed by Coles is extended in [19,20]. Moreover, Monty [21] proposed two different formulations for the wake law for pipe and channel flows.

The key feature of turbulence is instantaneous chaotic motion. The interaction of inner and outer regions is an intrinsically nonlinear process [22]. The turbulent flow is subjected to inertial and viscous forces acting with different intensities at different wall-normal positions. The internal turbulent flows are composed of recurrent and quantifiable coherent structures, having different length scales (the Reynolds number can be viewed as a measure of separation between the largest inertial and the smallest viscous scales). The need for adequate scale separation when considering coherent structures in pipe/channel turbulent flow is given in [22]. Interactions between these coherent structures are different in pipe and channel flows [23–25]. The turbulence, which is generated at the wall, is transported outward; the different available space in the pipe/channel core region causes different

turbulence behavior. In a channel, the space for turbulence to be transported and to be developed is constant along the wall-normal coordinate; in a pipe, this space is successively reduced to zero toward the center. In a pipe, this leads to more intense interactions between turbulent structures.

DNS data [21,26] and experiment [27–29] databases allow us to elucidate the statistics on the velocity fluctuations in pipe and channel flows. The streamwise turbulence intensities in the pipe and channel flows show no significant differences in the inner and outer layers; the wall-normal and spanwise turbulence intensities in pipe flow are larger than those of channel flow in the outer layer.

In this note, we provide a comparison over a range of $Re_\tau$ from 180 to 2000, where $Re_\tau = \Theta u_\tau / v$ is the friction Reynolds number, $v$ the kinematic viscosity, $\Theta$ the pipe radius or the channel half-height, $u_\tau = \sqrt{\tau_w/\rho}$ the friction velocity, $\tau_w$ the wall shear stress, and $\rho$ the fluid density. For channel flows, we deduce the MVP by using the Indirect Turbulence Model (ITM) proposed in [30]; for pipe flows, we extract the needed information from the Direct Numerical Simulation (DNS) database available in the open literature [26,31,32]. A one-to-one comparison of turbulent pipe and channel flows at the identical friction Reynolds numbers allows us to define the global indicators and to investigate the Reynolds number effects. This line of reasoning allows us to observe some deviations among the MVPs extracted from the two different pipe DNS databases. After some remarks on the ITM (Section 2), in Section 3 we illustrate and discuss the results of the comparison, and we highlight the discrepancies in global parameters for pipe flows. In Section 4 we summarize the findings quantitatively, and we give the concluding comments.

## 2. Indirect Turbulence Model (ITM)

In this section, we re-examine the ITM as proposed in [30], which allows us to derive the MPV for 2D turbulent channel flows in smooth walls. By assuming a hyperbolic trend of the turbulent shear stress, the mean velocity $u^+$ in the streamwise direction is expressed as a function of the wall-normal coordinate $y^+$ and of the friction Reynolds number $Re_\tau = y^+_{max}$, where the superscript $+$ stands for normalization with the inner variable, the friction velocity $u_\tau$, and the fluid kinematic viscosity $v$. According to the ITM, the relationship $u^+ = u^+(y^+, Re_\tau)$ is given as:

$$u^+ = (\varphi_1 + \psi_1/2)y^+/y^+_{max} + \varphi_2 \left(y^+/y^+_{max}\right)^2 + \varphi_3\psi_3 + \varphi_3 \ln \psi_2 \tag{1}$$

where:

$$\psi_1 = \sqrt{P_1\left(y^+/y^+_{max}\right)^2 + P_2 y^+/y^+_{max} + P_3} \tag{2}$$

$$\psi_2 = \left(P_2/\sqrt{P_1} + 2\sqrt{P_3}\right) / \left(\left(2P_1 y^+/y^+_{max} + P_2\right)/\sqrt{P_1} + 2\psi_1\right) \tag{3}$$

$$\psi_3 = \psi_1 - \sqrt{P_3} \tag{4}$$

$$\varphi_1 = Re_\tau + D \tag{5}$$

$$\varphi_2 = \left(B - y^+_{max}\right)/2 \tag{6}$$

$$\varphi_3 = \left(\sqrt{P_1}P_2\right) / \left(4P_1^{3/2}\right) \tag{7}$$

$$\varphi_4 = \left(P_2^2 - 4P_1 P_3\right) / \left(8P_1^{3/2}\right) \tag{8}$$

$$P_1 = B^2 - C \tag{9}$$

$$P_2 = 2(BD - E) \tag{10}$$

$$P_3 = D^2 + C + 2E \tag{11}$$

$$B = Re_\tau(1 - f_1) \tag{12}$$

$$C = \left(y^+_{max}{}^2 - By^+_{max}\right)/f_2 - y^+_{max}{}^2 + 2By^+_{max} \tag{13}$$

$$D = -1/y^+{}_{max}\left(\left(y^+{}_{max}{}^2 - By^+{}_{max}\right)/f_3 + y^+{}_{max}{}^2\right) \tag{14}$$

$$E = \left(y^+{}_{max}{}^2 - By^+{}_{max}\right)/f_4 + y^+{}_{max}{}^2 - By^+{}_{max} + Dy^+{}_{max} \tag{15}$$

$$f_1 = \left(3.655y^+{}_{max}{}^2 + 25{,}704.994y^+{}_{max} - 55{,}013.808\right)\cdot 10^{-6} \tag{16}$$

$$f_2 = \left(6.991y^+{}_{max}{}^2 + 39{,}476.172y^+{}_{max} - 2{,}873{,}405.419\right)\cdot 10^{-6} \tag{17}$$

$$f_3 = \left(-7.409y^+{}_{max}{}^2 - 49{,}231.626y^+{}_{max} + 556{,}178.423\right)\cdot 10^{-6} \tag{18}$$

$$f_4 = \left(-23.766y^+{}_{max}{}^2 - 82{,}908.798y^+{}_{max} + 4{,}325{,}049.776\right)\cdot 10^{-6} \tag{19}$$

We underline that the ITM appears as a generalization of the log-law in wall-bounded turbulent flows (in both models, the turbulent shear stress exhibits a hyperbolic trend). In comparison to the very simple structure of the log-law, the ITM provides a complex relationship $u^+ = u^+(y^+, Re_\tau)$; on the other hand, this relationship satisfies both the boundary condition at $y^+ = 0$, $u^+(y^+ = 0) = 0$, and the centerline condition at $y^+ = Re_\tau$, $\frac{\mathrm{d}}{\mathrm{d}y^+}u^+(y^+ = Re_\tau) = 0$ [30].

In Figure 1, we show the comparison of the MPV given by Equation (1) versus the channel DNS data [33–38] (the corresponding information is given in Table 1).

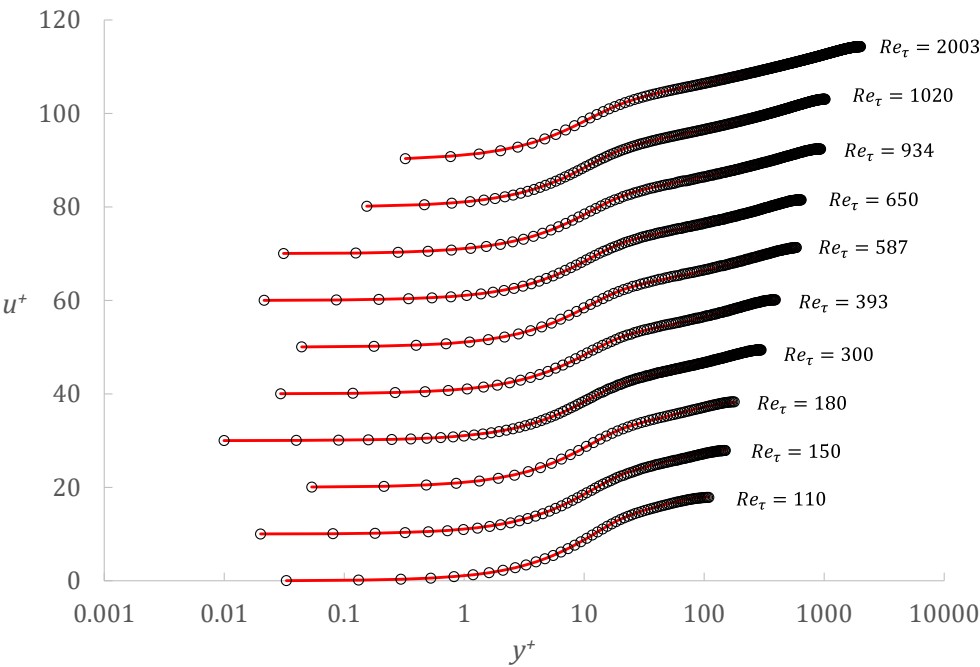

**Figure 1.** MVP; ○ DNS data; and − ITM. The plots are shifted vertically by 10 units.

**Table 1.** $Re_\tau$, $y^+{}_{max}$, and Hellinger distance $He$.

| $Re_\tau$ | $y^+{}_{max}$ | $HD$ |
|---|---|---|
| 110 | 109.43 | $1.21\cdot 10^{-1}$ |
| 150 | 150.00 | $3.20\cdot 10^{-1}$ |
| 180 | 178.12 | $1.39\cdot 10^{-1}$ |
| 300 | 298.00 | $3.66\cdot 10^{-1}$ |
| 393 | 392.00 | $2.84\cdot 10^{-1}$ |
| 587 | 587.00 | $2.61\cdot 10^{-1}$ |
| 650 | 642.54 | $2.72\cdot 10^{-1}$ |
| 934 | 933.96 | $2.83\cdot 10^{-1}$ |
| 1020 | 1016.36 | $3.93\cdot 10^{-1}$ |
| 2003 | 2004.30 | $2.37\cdot 10^{-1}$ |

This comparison shows a very good performance of the ITM. As a metric to measure the match between the DNS data and the ITM data, we use the Hellinger distance *HD*, given as [39]:

$$HD\left(u^+{}_{DNS}||u^+{}_{ITM}\right) = \sqrt{2\left(\sum_{y^+=0}^{Re_\tau}\left(\left(u^+{}_{DNS}(y^+)\right)^{0.5} - \left(u^+{}_{ITM}(y^+)\right)^{0.5}\right)^2\right)} \qquad (20)$$

For all cases, *HD* is very close to zero (see Table 1).

In Figures 2–6, we compare the global indicators extracted from the DNS database and from the ITM (details are given in Table 2).

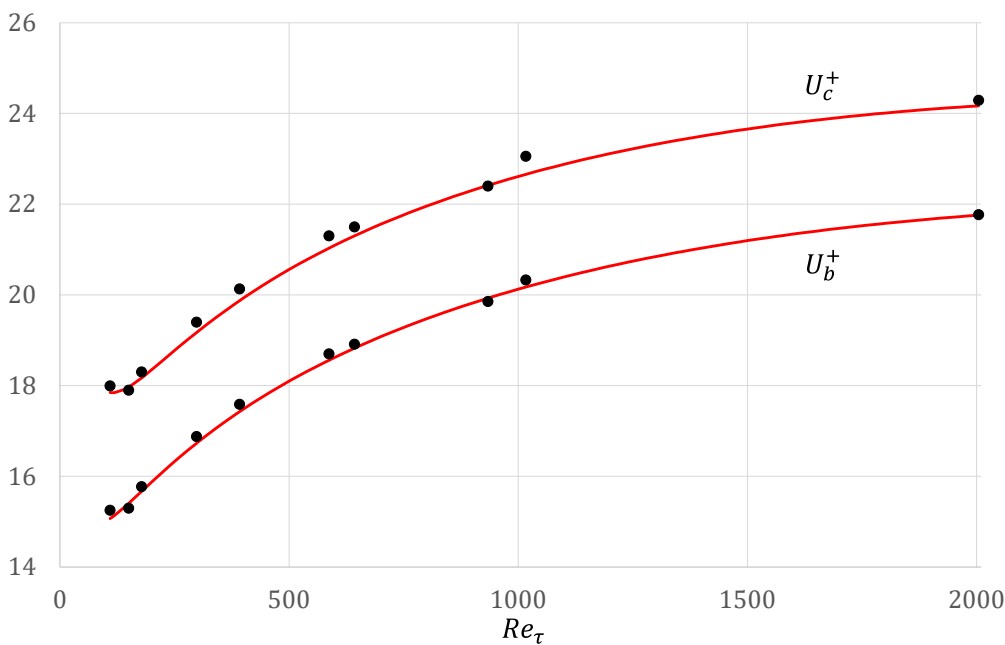

**Figure 2.** Trend of the mean velocity $U_b{}^+$ and of the centerline velocity $U_c{}^+$; ● DNS data; and − ITM.

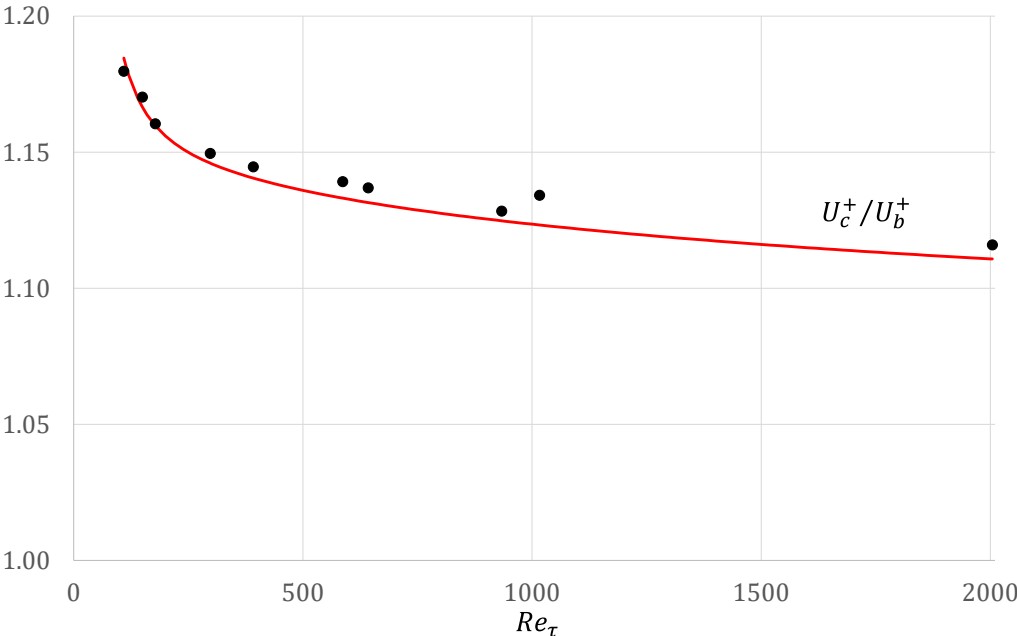

**Figure 3.** Trend of the ratio $U_c{}^+/U_b{}^+$; ● DNS data; and − ITM.

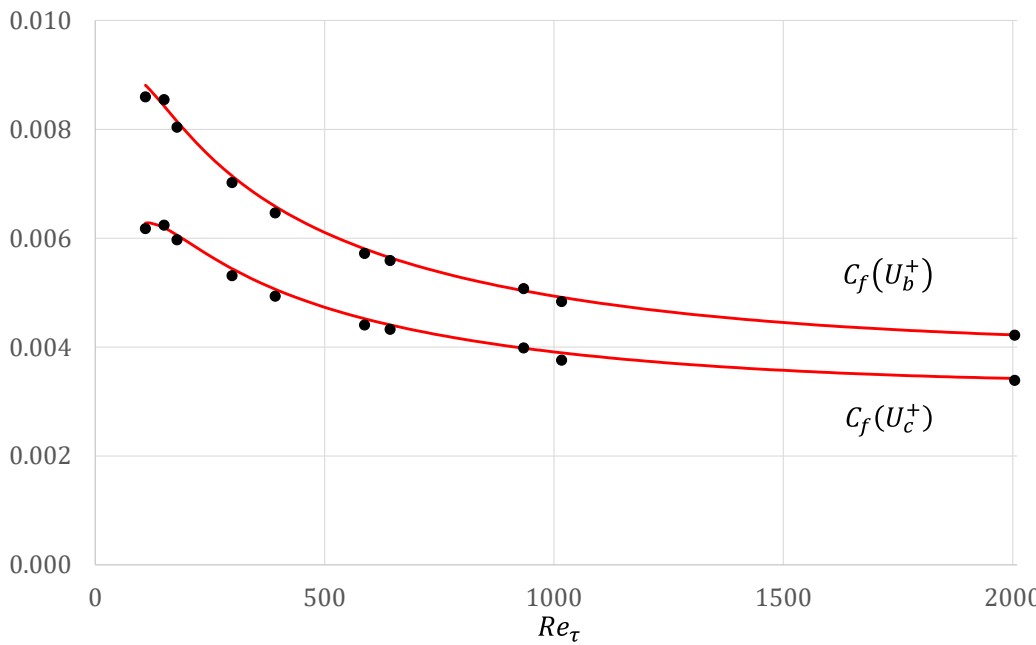

**Figure 4.** Trend of the skin friction based on the mean velocity $C_f(U_b{}^+) = 2/(U_b{}^+)^2$ and of the skin friction based on the centerline velocity $C_f(U_c{}^+) = 2/(U_c{}^+)^2$; $\bullet$ DNS data; and $-$ ITM.

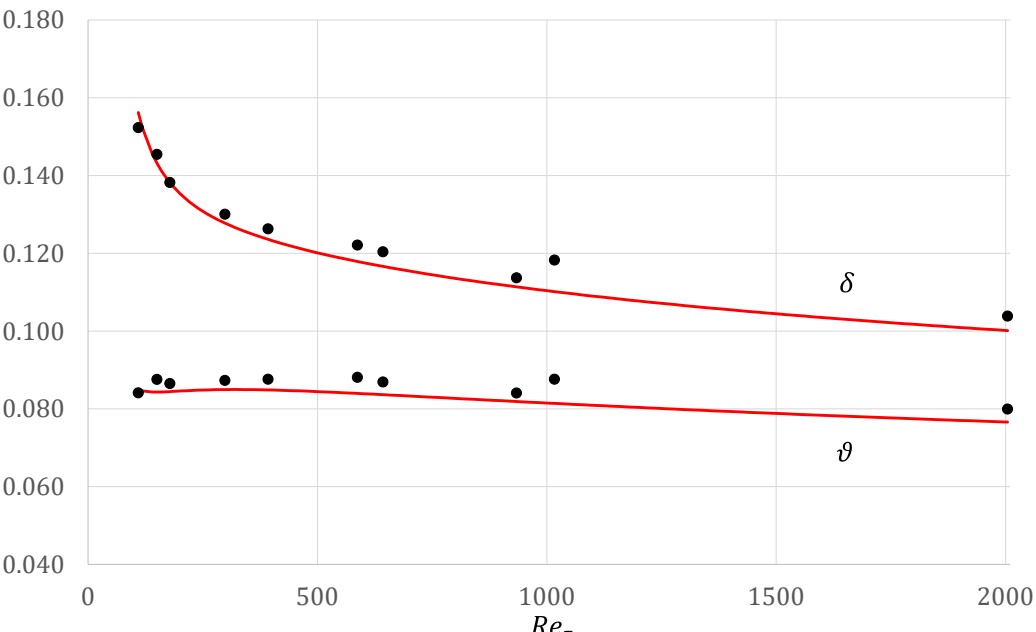

**Figure 5.** Trend of the displacement thickness $\delta = \int_0^1 \left(1 - \frac{u^+}{U_c{}^+}\right) d\zeta$, where $\zeta = y^+/y^+{}_{max}$, and of the momentum thickness $\vartheta = \int_0^1 \frac{u^+}{U_c{}^+}\left(1 - \frac{u^+}{U_c{}^+}\right) d\zeta$; $\bullet$ DNS data; and $-$ ITM.

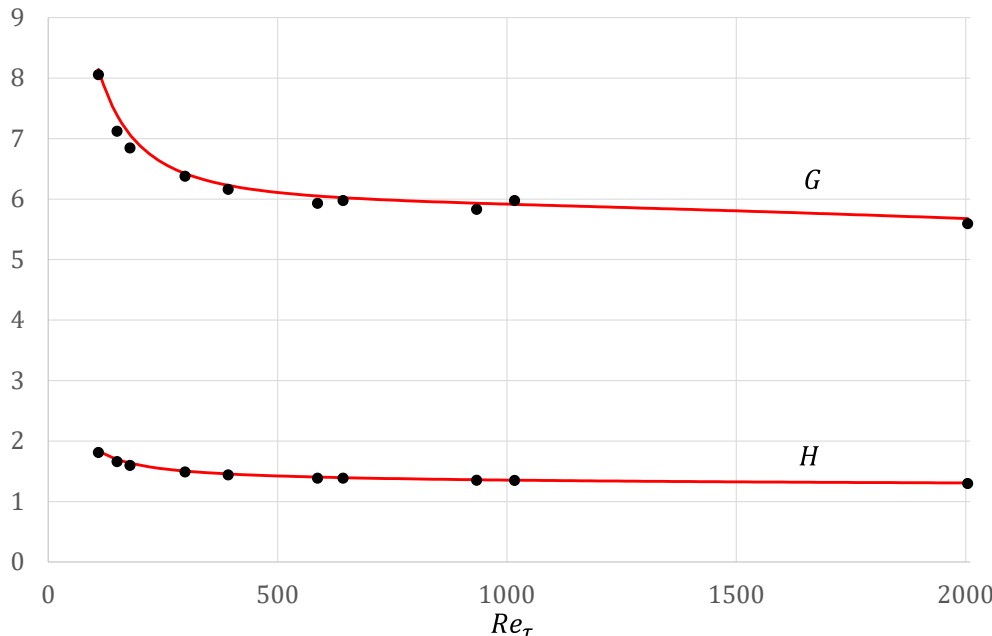

**Figure 6.** Trend of the first shape factor $H = \delta/\vartheta$ and of the second shape factor $G = U_c^+((H-1)/H)$; ● DNS data; and — ITM.

**Table 2.** Mean flow properties extracted from DNS databases and from ITM. The parameters are defined as: $U_b^+$ is the mean velocity; $U_c^+$ the centerline velocity; $C_f(U_b^+)$ the skin friction based on the mean velocity $C_f(U_b^+) = 2/U_b^{+2}$; $C_f(U_c^+)$ the skin friction based on the centerline velocity $C_f(U_c^+) = 2/U_c^{+2}$; $\delta$ the displacement thickness $\delta = \int_0^1 \left(1 - \frac{u^+}{U_c^+}\right)\mathrm{d}\zeta$, where $\zeta = y^+/y^+_{max}$; $\vartheta$ the momentum thickness $\vartheta = \int_0^1 \frac{u^+}{U_c^+}\left(1 - \frac{u^+}{U_c^+}\right)\mathrm{d}\zeta$; $H$ the first shape factor $H = \delta/\vartheta$; and $G$ the second shape factor $G = U_c^+((H-1)/H)$.

| Database/ Error | $Re_\tau$ | $y^+_{max}$ | $U_b^+$ | $U_c^+$ | $U_c^+/U_b^+$ | $C_f(U_b^+)$ | $C_f(U_c^+)$ | $\delta$ | $\vartheta$ | $H$ | $G$ |
|---|---|---|---|---|---|---|---|---|---|---|---|
| DNS | 110 | 109.43 | 15.25 | 17.99 | 1.18 | $8.60\cdot10^{-3}$ | $6.18\cdot10^{-3}$ | $1.52\cdot10^{-1}$ | $8.41\cdot10^{-2}$ | 1.81 | 8.06 |
| | 150 | 150.00 | 15.30 | 17.90 | 1.17 | $8.55\cdot10^{-3}$ | $6.24\cdot10^{-3}$ | $1.45\cdot10^{-1}$ | $8.76\cdot10^{-2}$ | 1.66 | 7.12 |
| | 180 | 178.12 | 15.77 | 18.30 | 1.16 | $8.04\cdot10^{-3}$ | $5.97\cdot10^{-3}$ | $1.38\cdot10^{-1}$ | $8.65\cdot10^{-1}$ | 1.60 | 6.85 |
| | 300 | 298.00 | 16.88 | 19.40 | 1.15 | $7.02\cdot10^{-3}$ | $5.31\cdot10^{-3}$ | $1.30\cdot10^{-1}$ | $8.73\cdot10^{-2}$ | 1.49 | 6.38 |
| | 393 | 392.00 | 17.59 | 20.13 | 1.14 | $6.47\cdot10^{-3}$ | $4.94\cdot10^{-3}$ | $1.26\cdot10^{-1}$ | $8.77\cdot10^{-2}$ | 1.44 | 6.16 |
| | 587 | 587.00 | 18.70 | 21.30 | 1.14 | $5.72\cdot10^{-3}$ | $4.41\cdot10^{-3}$ | $1.22\cdot10^{-1}$ | $8.81\cdot10^{-2}$ | 1.39 | 5.93 |
| | 650 | 642.54 | 18.91 | 21.50 | 1.14 | $5.59\cdot10^{-3}$ | $4.33\cdot10^{-3}$ | $1.20\cdot10^{-1}$ | $8.69\cdot10^{-2}$ | 1.38 | 5.98 |
| | 934 | 933.96 | 19.85 | 22.40 | 1.13 | $5.07\cdot10^{-3}$ | $3.99\cdot10^{-3}$ | $1.14\cdot10^{-1}$ | $8.41\cdot10^{-2}$ | 1.35 | 5.83 |
| | 1020 | 1016.36 | 20.33 | 23.06 | 1.13 | $4.84\cdot10^{-3}$ | $3.76\cdot10^{-3}$ | $1.18\cdot10^{-1}$ | $8.76\cdot10^{-2}$ | 1.35 | 5.98 |
| | 2003 | 2004.30 | 21.77 | 24.29 | 1.12 | $4.22\cdot10^{-3}$ | $3.39\cdot10^{-3}$ | $1.04\cdot10^{-1}$ | $7.99\cdot10^{-2}$ | 1.30 | 5.59 |
| ITM | 110 | 109.43 | 15.16 | 17.85 | 1.18 | $8.71\cdot10^{-3}$ | $6.28\cdot10^{-3}$ | $1.51\cdot10^{-1}$ | $8.33\cdot10^{-2}$ | 1.81 | 7.99 |
| | 150 | 150.00 | 15.46 | 17.97 | 1.16 | $8.37\cdot10^{-3}$ | $6.19\cdot10^{-3}$ | $1.40\cdot10^{-1}$ | $8.32\cdot10^{-2}$ | 1.68 | 7.27 |
| | 180 | 178.12 | 15.75 | 18.17 | 1.15 | $8.07\cdot10^{-3}$ | $6.06\cdot10^{-3}$ | $1.33\cdot10^{-1}$ | $8.28\cdot10^{-2}$ | 1.61 | 6.89 |
| | 300 | 298.00 | 16.75 | 19.17 | 1.14 | $7.13\cdot10^{-3}$ | $5.44\cdot10^{-3}$ | $1.26\cdot10^{-1}$ | $8.44\cdot10^{-2}$ | 1.50 | 6.36 |
| | 393 | 392.00 | 17.46 | 19.87 | 1.14 | $6.56\cdot10^{-3}$ | $5.06\cdot10^{-3}$ | $1.21\cdot10^{-1}$ | $8.40\cdot10^{-2}$ | 1.45 | 6.13 |
| | 587 | 587.00 | 18.59 | 21.03 | 1.13 | $5.79\cdot10^{-3}$ | $4.52\cdot10^{-3}$ | $1.16\cdot10^{-1}$ | $8.31\cdot10^{-2}$ | 1.40 | 5.95 |
| | 650 | 642.54 | 18.84 | 21.30 | 1.13 | $5.63\cdot10^{-3}$ | $4.41\cdot10^{-3}$ | $1.15\cdot10^{-1}$ | $8.31\cdot10^{-2}$ | 1.39 | 5.96 |
| | 934 | 933.96 | 19.94 | 22.41 | 1.12 | $5.03\cdot10^{-3}$ | $3.98\cdot10^{-3}$ | $1.10\cdot10^{-1}$ | $8.13\cdot10^{-2}$ | 1.35 | 5.87 |
| | 1020 | 1016.36 | 20.18 | 22.66 | 1.12 | $4.91\cdot10^{-3}$ | $3.90\cdot10^{-3}$ | $1.09\cdot10^{-1}$ | $8.09\cdot10^{-2}$ | 1.35 | 5.87 |
| | 2003 | 2004.30 | 21.76 | 24.17 | 1.11 | $4.22\cdot10^{-3}$ | $3.42\cdot10^{-3}$ | $9.94\cdot10^{-2}$ | $7.63\cdot10^{-2}$ | 1.30 | 5.63 |

**Table 2.** *Cont.*

| Database/ Error | $Re_\tau$ | $y^+_{max}$ | $U_b^+$ | $U_c^+$ | $U_c^+/U_b^+$ | $C_f(U_b^+)$ | $C_f(U_c^+)$ | $\delta$ | $\vartheta$ | $H$ | $G$ |
|---|---|---|---|---|---|---|---|---|---|---|---|
| | | | 0.62% | 0.80% | 0.18% | 1.26% | 1.63% | 1.01% | 0.95% | 0.06% | 0.88% |
| | | | 1.04% | 0.38% | 0.66% | 2.06% | 0.75% | 3.90% | 4.99% | 1.15% | 2.11% |
| | | | 0.16% | 0.73% | 0.57% | 0.32% | 1.47% | 3.56% | 4.32% | 0.79% | 0.57% |
| | | | 0.76% | 1.19% | 0.43% | 1.54% | 2.42% | 2.90% | 3.37% | 0.49% | 0.21% |
| relative error | | | 0.73% | 1.28% | 0.55% | 1.48% | 2.61% | 3.83% | 4.18% | 0.36% | 0.47% |
| | | | 0.58% | 1.28% | 0.70% | 1.16% | 2.60% | 5.09% | 5.72% | 0.67% | 0.43% |
| | | | 0.37% | 0.94% | 0.57% | 0.74% | 1.90% | 4.21% | 4.46% | 0.27% | 0.25% |
| | | | 0.46% | 0.06% | 0.40% | 0.92% | 0.12% | 3.14% | 3.34% | 0.21% | 0.67% |
| | | | 0.72% | 1.74% | 1.03% | 1.45% | 3.56% | 7.73% | 7.72% | 0.01% | 1.77% |
| | | | 0.02% | 0.51% | 0.49% | 0.04% | 1.03% | 4.26% | 4.56% | 0.32% | 0.56% |
| medium relative error | | | 0.55% | 0.89% | 0.56% | 1.10% | 1.81% | 3.96% | 4.36% | 0.43% | 0.79% |
| maximum relative error | | | 1.04% | 1.74% | 1.03% | 2.06% | 3.56% | 7.73% | 7.72% | 1.15% | 2.11% |

Figure 2 shows that the ITM provides a good fit for the mean velocity $U_b^+$ (the percentual error is almost always less than 1%), while it underestimates the centerline velocity $U_c^+$ (although the maximum error is less than 2%).

Figure 3 shows that the underestimation of $U_c^+$ in the ITM is reflected in an underestimation of the ratio $U_c^+/U_b^+$ (although the percentual error is almost always less than 1%), while Figure 4 shows an acceptable fit in both $C_f(U_b^+)$ and $C_f(U_c^+)$.

Figure 5 shows that the underestimation of $U_c^+$ in the ITM is reflected in an underestimation in both parameters $\delta$ and $\vartheta$, while Figure 5 shows an acceptable fit in both parameters $G$ and $H$.

The ITM provides an accurate estimation for $U_b^+$, while the $U_c^+$ is almost always underestimated. As a consequence, some parameters, such as $\delta$ and $\vartheta$, suffer from underestimation. From a general point of view, the obtained results show the reliability of the ITM to reproduce the global indicators of the turbulent channel flow.

## 3. Channel VS Pipe

In this section, we provide a comparison between the mean flow properties extracted from the pipe DNS database [26,31,32] and those deduced by the ITM. In Figure 7, we show the comparison of the MPVs. In Table 3, we give the corresponding information.

**Table 3.** $Re_\tau$ and $y^+_{max}$.

| Database | $Re_\tau$ | $y^+_{max}$ |
|---|---|---|
| | 180 | 172.30 |
| [26] | 500 | 500.25 |
| | 1000 | 1001.92 |
| | 2000 | 2003.26 |
| | 180 | 181.89 |
| [31,32] | 500 | 495.26 |
| | 1000 | 1136.59 |
| | 2000 | 1977.24 |

In reference to the pipe DNS database [26] (Figure 7), the comparison confirms previous results which display that in the inner layer, the MVPs are essentially indistinguishable; in the outer layer, there are remarkable differences, principally in the wake layer. As stated in Section 2, the different available spaces in the pipe/channel core region causes different turbulent behavior: the space limitations in a pipe lead to more intense interactions between turbulent structures, with an increment of the streamwise mean velocity $u^+$. In reference to the pipe DNS database [31,32] (Figure 8), the MPVs present differences in both the inner and outer layers, and in the wake layer, these differences become important (as expected). We find other discordant results that can be attributed to a dissimilar performance of the

numerical schemes used in [26,31,32] when we compare the global indicators for pipe flow. Details are given in Table 4 and in Figures 9–17.

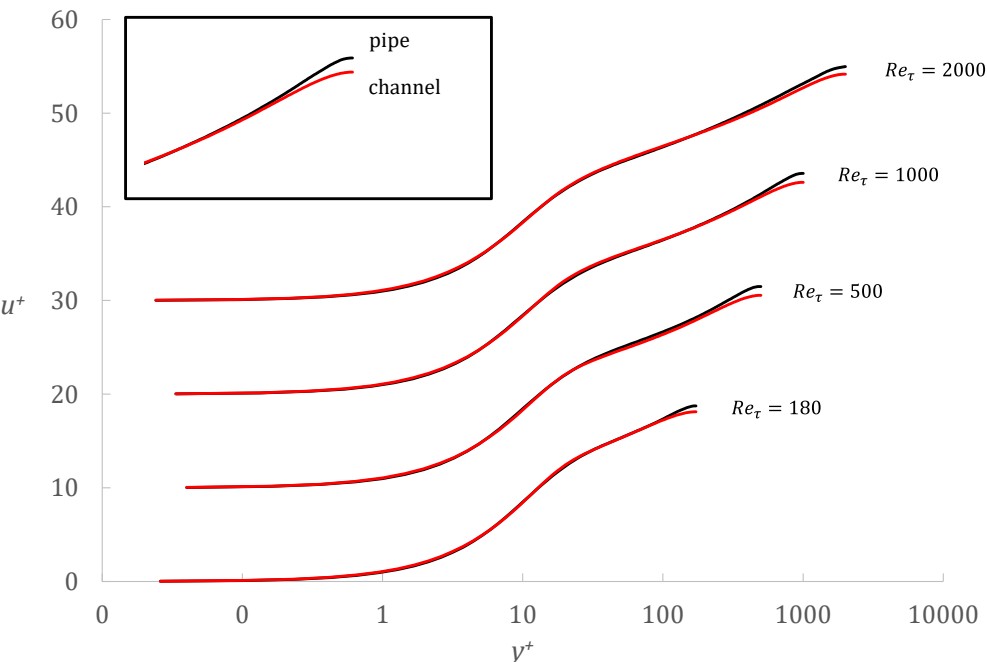

**Figure 7.** MVP; – pipe DNS database [26]; and – channel ITM model. The plots are shifted vertically by 10 units.

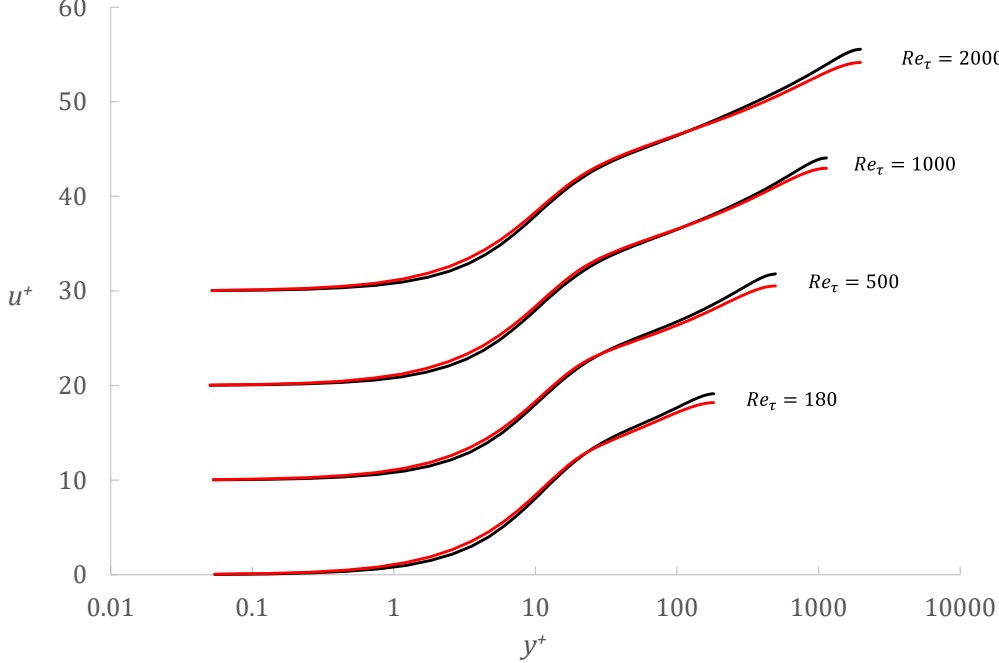

**Figure 8.** MVP; – DNS database [31,32]; and – channel ITM model. The plots are shifted vertically by 10 units.

**Table 4.** Mean flow properties extracted from pipe DNS database [26,31,32] and from channel ITM. The parameters are defined as: $U_b{}^+$ is the mean velocity; $U_c{}^+$ the centerline velocity; $C_f(U_b{}^+)$ the skin friction based on the mean velocity $C_f(U_b{}^+) = 2/U_b{}^{+2}$; $C_f(U_c{}^+)$ the skin friction based on the centerline velocity $C_f(U_c{}^+) = 2/U_c{}^{+2}$; $\delta$ the displacement thickness, which for pipe is defined as $\delta(2-\delta) = 2\int_0^1 \left(1 - \frac{u^+}{U_c{}^+}\right)(1-\zeta)d\zeta$, where $\zeta = y^+/y^+{}_{max}$; $\vartheta$ the momentum thickness $\vartheta(2-\vartheta) = 2\int_0^1 \frac{u^+}{U_c{}^+}\left(1 - \frac{u^+}{U_c{}^+}\right)(1-\zeta)d\zeta$; $H$ the first shape factor $H = \delta/\vartheta$; and $G$ the second shape factor $G = U_c{}^+((H-1)/H)$.

| Database | $Re_\tau$ | $y^+{}_{max}$ | $U_b{}^+$ | $U_c{}^+$ | $U_c{}^+/U_b{}^+$ | $C_f(U_b{}^+)$ | $C_f(U_c{}^+)$ | $\delta$ | $\vartheta$ | $H$ | $G$ |
|---|---|---|---|---|---|---|---|---|---|---|---|
| [26] | 180 | 172.30 | 13.96 | 18.75 | 1.34 | $1.03\cdot10^{-2}$ | $5.69\cdot10^{-3}$ | $2.42\cdot10^{-1}$ | $1.05\cdot10^{-1}$ | 2.31 | 10.63 |
| | 500 | 500.25 | 16.87 | 21.50 | 1.27 | $7.03\cdot10^{-3}$ | $4.33\cdot10^{-3}$ | $2.05\cdot10^{-1}$ | $1.05\cdot10^{-1}$ | 1.95 | 10.50 |
| | 1000 | 1001.92 | 18.78 | 23.57 | 1.26 | $5.67\cdot10^{-3}$ | $3.60\cdot10^{-3}$ | $1.97\cdot10^{-1}$ | $1.06\cdot10^{-2}$ | 1.86 | 10.87 |
| | 2000 | 2003.26 | 20.61 | 24.96 | 1.21 | $4.71\cdot10^{-3}$ | $3.21\cdot10^{-3}$ | $1.71\cdot10^{-1}$ | $9.78\cdot10^{-2}$ | 1.74 | 10.65 |
| [31,32] | 180 | 181.89 | 14.27 | 19.14 | 1.34 | $9.82\cdot10^{-3}$ | $5.46\cdot10^{-3}$ | $2.37\cdot10^{-1}$ | $1.02\cdot10^{-1}$ | 2.33 | 10.91 |
| | 500 | 495.26 | 17.01 | 21.81 | 1.28 | $6.91\cdot10^{-3}$ | $4.20\cdot10^{-3}$ | $2.12\cdot10^{-1}$ | $1.04\cdot10^{-1}$ | 2.04 | 11.14 |
| | 1000 | 1136.59 | 19.27 | 24.07 | 1.25 | $5.39\cdot10^{-3}$ | $3.45\cdot10^{-3}$ | $1.95\cdot10^{-1}$ | $1.06\cdot10^{-1}$ | 1.85 | 11.05 |
| | 2000 | 1977.24 | 20.80 | 25.55 | 1.23 | $4.62\cdot10^{-3}$ | $3.06\cdot10^{-3}$ | $1.83\cdot10^{-1}$ | $1.03\cdot10^{-1}$ | 1.78 | 11.22 |
| ITM | 180 | 172.30 | 15.67 | 18.12 | 1.16 | $8.15\cdot10^{-3}$ | $6.09\cdot10^{-3}$ | $1.36\cdot10^{-1}$ | $8.32\cdot10^{-2}$ | 1.63 | 7.00 |
| | 180 | 181.89 | 15.79 | 18.20 | 1.15 | $8.03\cdot10^{-3}$ | $6.04\cdot10^{-3}$ | $1.33\cdot10^{-1}$ | $8.30\cdot10^{-2}$ | 1.60 | 6.81 |
| | 500 | 495.26 | 18.10 | 20.53 | 1.13 | $6.10\cdot10^{-3}$ | $4.75\cdot10^{-3}$ | $1.18\cdot10^{-1}$ | $8.38\cdot10^{-2}$ | 1.41 | 5.97 |
| | 500 | 500.25 | 18.13 | 20.56 | 1.13 | $6.08\cdot10^{-3}$ | $4.73\cdot10^{-3}$ | $1.18\cdot10^{-1}$ | $8.35\cdot10^{-2}$ | 1.41 | 6.01 |
| | 1000 | 1001.92 | 20.15 | 22.62 | 1.12 | $4.93\cdot10^{-3}$ | $3.91\cdot10^{-3}$ | $1.09\cdot10^{-1}$ | $8.09\cdot10^{-2}$ | 1.35 | 5.85 |
| | 1000 | 1136.59 | 20.50 | 22.97 | 1.12 | $4.76\cdot10^{-3}$ | $3.79\cdot10^{-3}$ | $1.08\cdot10^{-1}$ | $8.04\cdot10^{-2}$ | 1.34 | 5.80 |
| | 2000 | 1977.24 | 21.74 | 24.15 | 1.11 | $4.23\cdot10^{-3}$ | $3.43\cdot10^{-3}$ | $9.97\cdot10^{-2}$ | $7.65\cdot10^{-2}$ | 1.30 | 5.63 |
| | 2000 | 2003.26 | 21.76 | 24.17 | 1.11 | $4.22\cdot10^{-3}$ | $3.42\cdot10^{-3}$ | $9.95\cdot10^{-2}$ | $7.63\cdot10^{-2}$ | 1.30 | 5.64 |

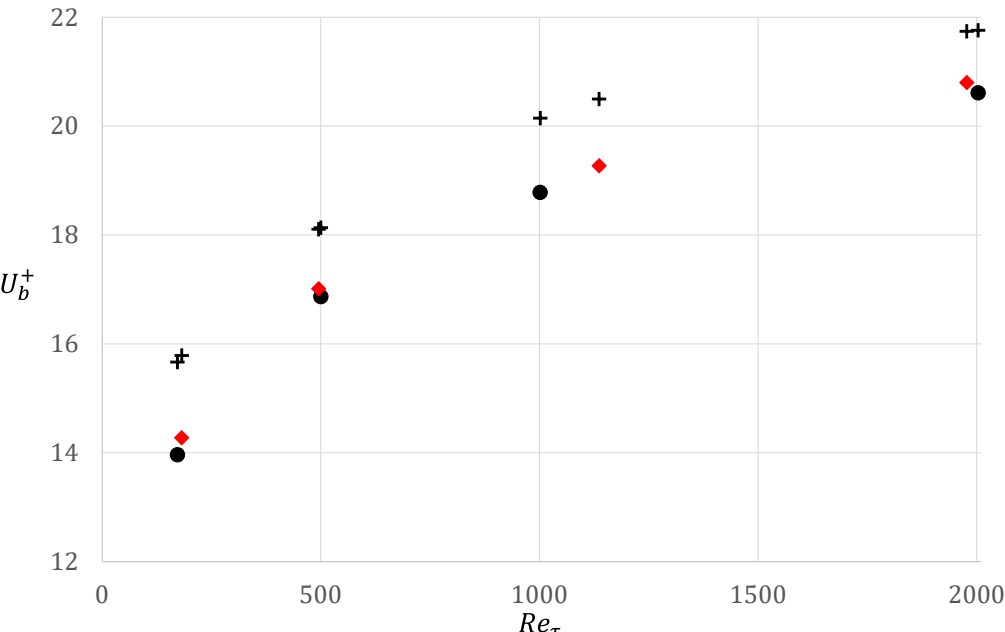

**Figure 9.** Trend of the mean velocity $U_b{}^+$; ● pipe DNS data [26]; ♦ pipe DNS data [31,32]; and + channel ITM.

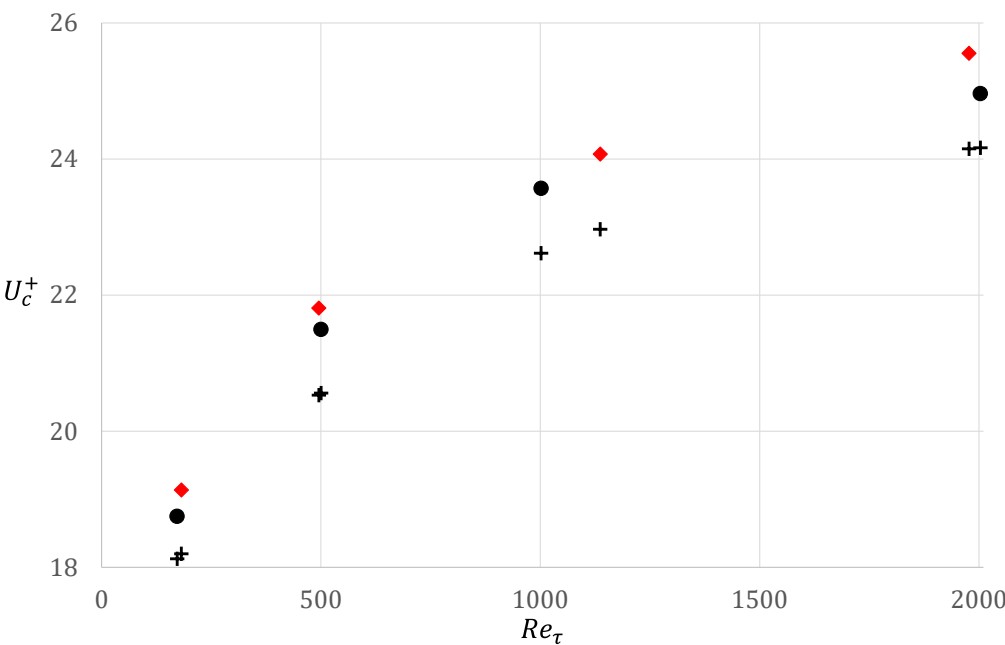

**Figure 10.** Trend of the centerline velocity $U_c{}^+$; ● pipe DNS data [26]; ◆ pipe DNS data [31,32]; and + channel ITM.

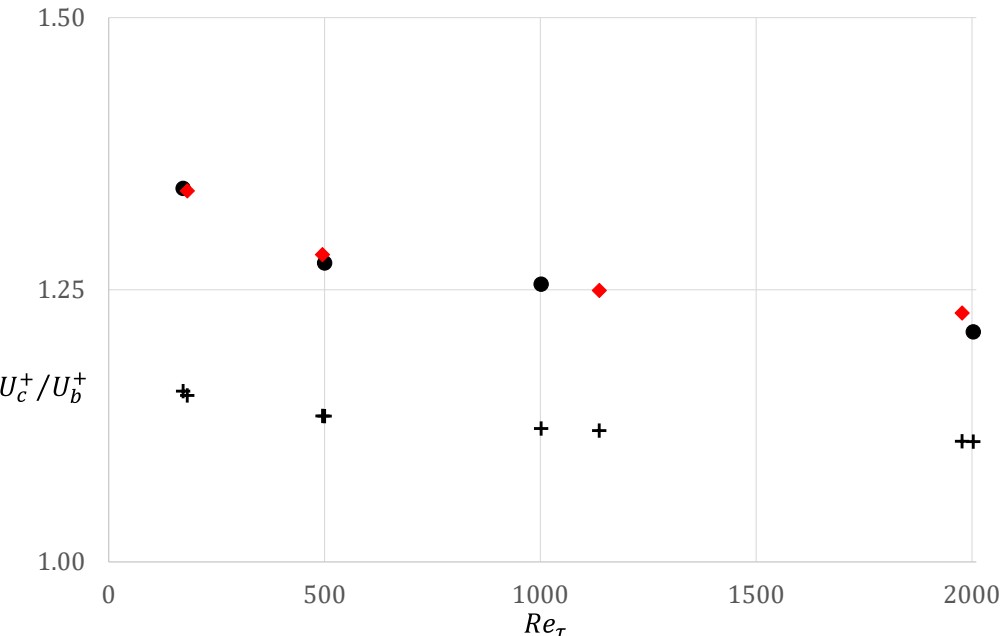

**Figure 11.** Trend of the ratio $U_c{}^+/U_b{}^+$; ● pipe DNS data [26]; ◆ pipe DNS data [31,32]; and + channel ITM.

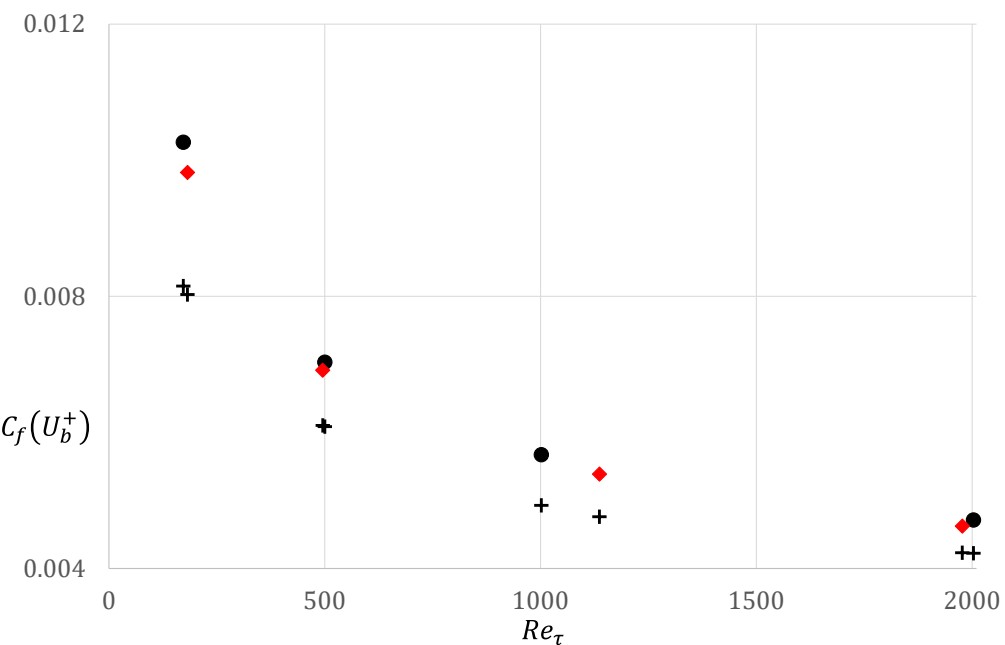

**Figure 12.** $C_f\left(U_b{}^+\right)$; • pipe DNS data [26]; ♦ pipe DNS data [31,32]; and + channel ITM.

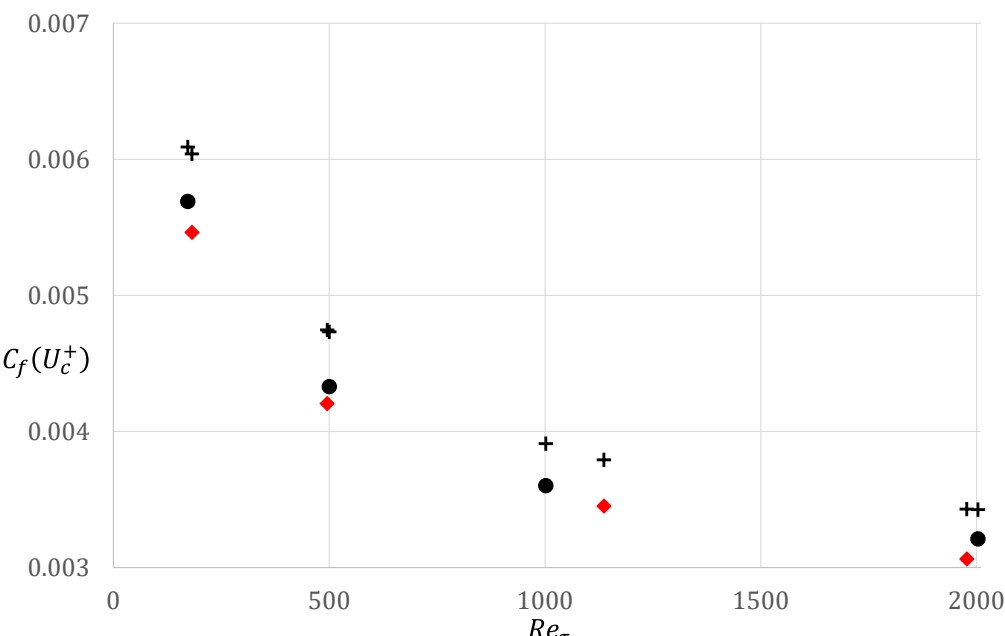

**Figure 13.** Trend of the skin friction $C_f\left(U_c{}^+\right)$; • pipe DNS data [26]; ♦ pipe DNS data [31,32]; and + channel ITM.

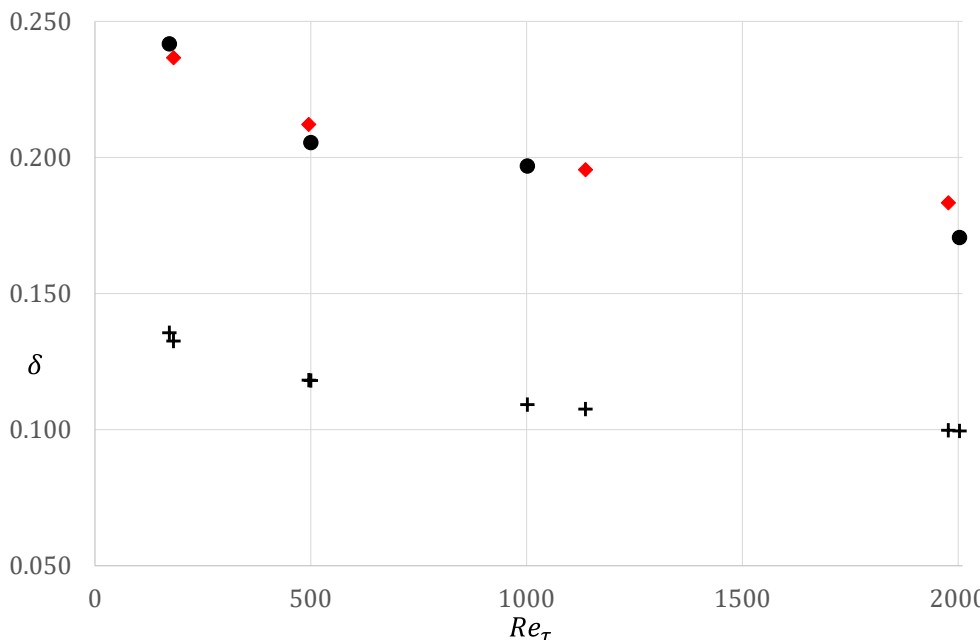

**Figure 14.** Trend of the displacement thickness $\delta$; • pipe DNS data [26]; ♦ pipe DNS data [31,32]; and + channel ITM.

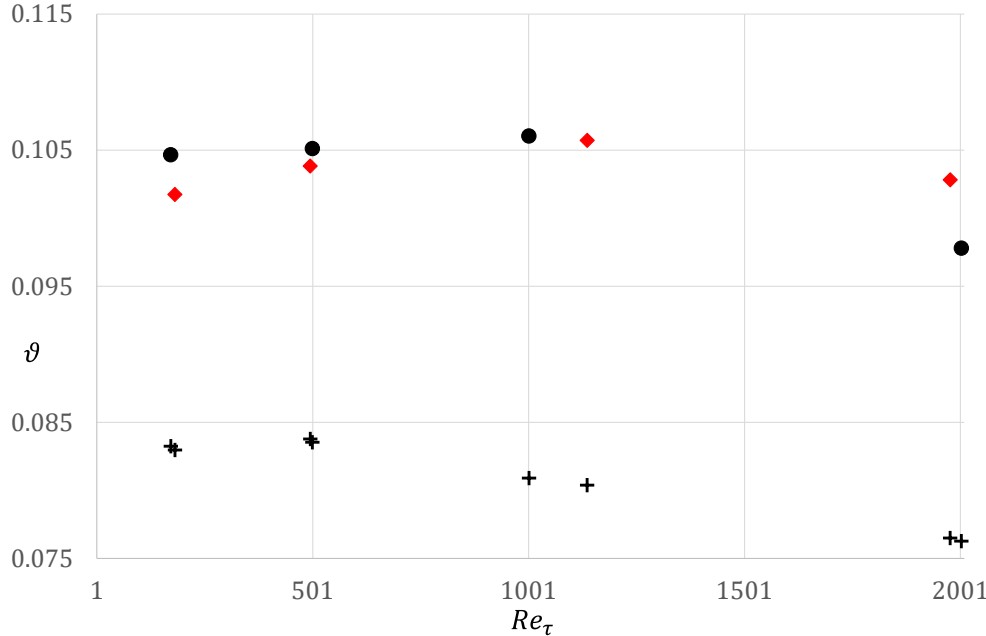

**Figure 15.** Trend of the momentum thickness $\vartheta$; • pipe DNS data [26]; ♦ pipe DNS data [31,32]; and + channel ITM.

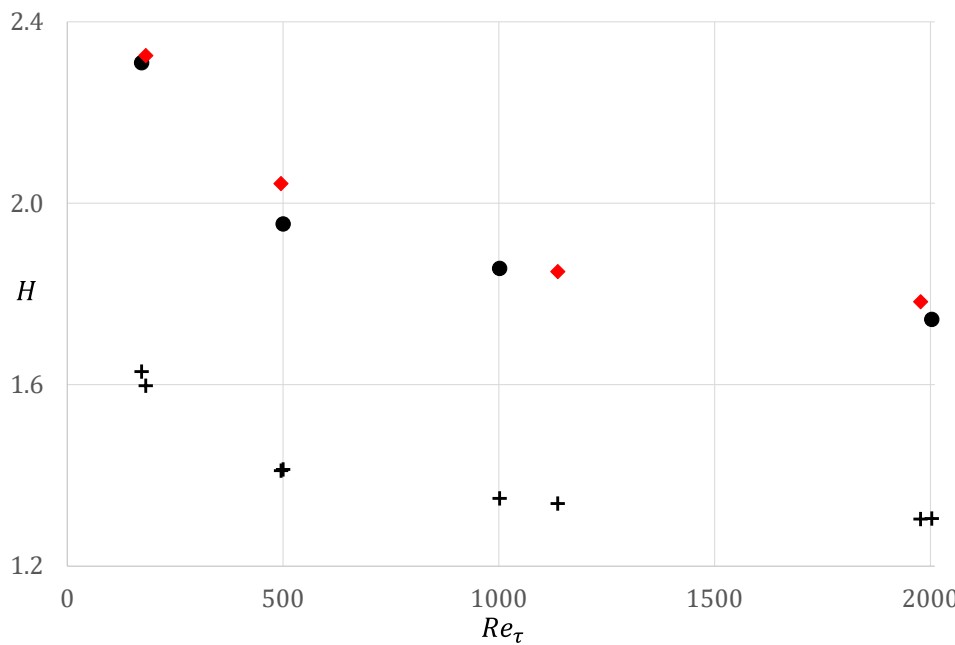

**Figure 16.** Trend of the first shape factor *H*; ● pipe DNS data [26]; ◆ pipe DNS data [31,32]; and + channel ITM.

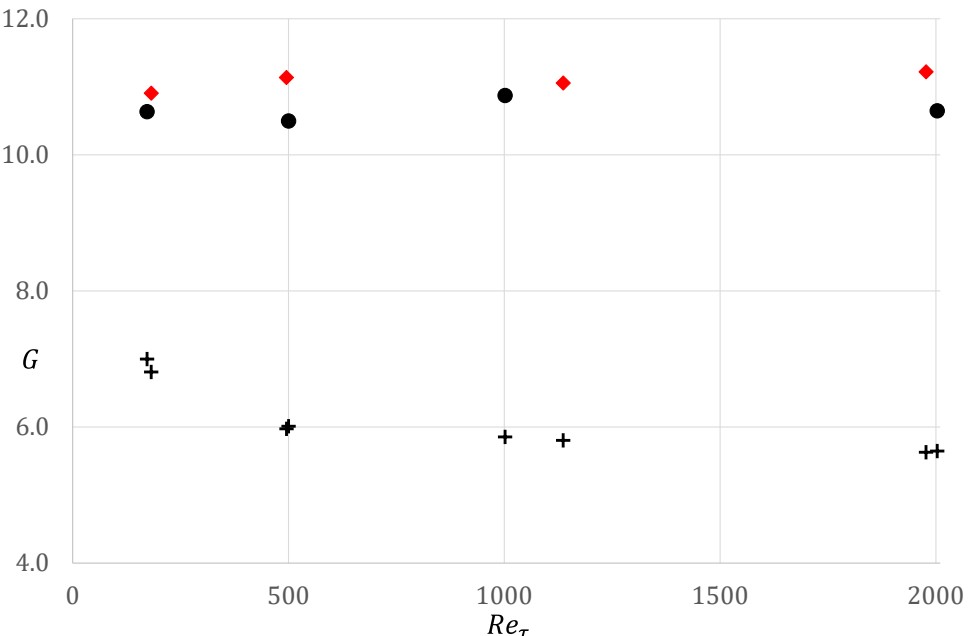

**Figure 17.** Trend of the second shape factor *G*; ● pipe DNS data [26]; ◆ pipe DNS data [31,32]; and + channel ITM.

As stated earlier, the space limitation in a pipe produces an increment in the centerline velocity $U_c^+$; on the other hand, due to the decrease in the cross-section area with increasing $y^+$, the mean velocity $U_b^+$ of the pipe flow is smaller than that of the channel flow.

The mean velocities $U_b^+$ obtained from the two different pipe DNS databases are in good agreement with each other (Figure 9); at the opposite, the centerline velocities $U_c^+$ extracted from the pipe DNS data [26] appear smaller than those extracted from the other pipe DNS data [31,32] (Figure 10). The other global parameters are, more or less severely, affected by these differences (Figures 11–17).

The obtained results confirm that the global indicators of pipe and channel flows are different. The mean velocity $U_b{}^+$ is larger in channel flow than in pipe flow, which determines $C_f(U_b{}^+)$ to be smaller; on the other hand, the centerline velocity $U_c{}^+$ is smaller in channel flow than in pipe flow, which causes $C_f(U_c{}^+)$ to be larger. The ratio $U_c{}^+/U_b{}^+$ for pipe flow exceeds the values in the channel flow; similar trends can be observed for the mean flow properties $\delta$, $\vartheta$, $H$, and $G$.

On the other hand, our results allow us to observe deviations in MVP obtained from the pipe DNS database [26,31,32]. These differences are reflected in global parameters: the mean velocity $U_b{}^+$ and the centerline velocity $U_c{}^+$ in [26] are smaller than in [31,32]; as a consequence, the respective skin frictions in [26] are larger than in [31,32]. Different trends also concern the parameters $\delta$, $\vartheta$, $H$, and $G$. For an accurate comparison, ideally, the datasets should consist of very similar Reynolds numbers and numerical parameters (i.e., temporal/spatial resolutions).

Finally, we notice that the data/curves in Figures 3–6, 9–14 and 16 seem to present an asymptotic behavior for $Re_\tau > 2000$ which could be related to the asymptotic behavior shown for the two coefficients of the eddy viscosity analytical model [40]. For large values of $Re_\tau > 2000$, the two coefficients of the analytical model reach asymptotic values equal, respectively, to $C_\alpha = 0.477$ and $C_1 = 2.17$. This will require further investigations in our future work.

## 4. Findings and Conclusions

In this brief note, we investigated the mean velocity properties of turbulent pipe and channel flows at low-to-moderate Reynolds numbers. We provided a one-to-one comparison at identical Reynolds numbers: for pipe flows, we extracted the needed information from DNS databases available in the open literature [26,31,32]; for channel flows, we used the ITM proposed in [30]. After some remarks on the ITM and on the reliability of this model to reproduce the global indicator, we examined the differences between fully developed flows in pipes and channels.

Preliminarily, we observed some deviations between the MVPs obtained from the pipe DNS databases [26,31,32]. These differences are reflected on global parameters: the mean velocity $U_b{}^+$ in [26] is about 1.5% smaller than in [31,32]; the centerline velocity $U_c{}^+$ in [26] is about 2% smaller than in [31,32]; the skin friction based on the mean velocity in [26] is about 3% larger than in [31,32]; and the skin friction based on the centerline velocity in [26] is about 4% larger than in [31,32]. These discrepancies, which can be due to a dissimilar performance of the numerical schemes used in [26,31,32], should lead to a reconsideration of the fidelity of the DNS data.

The comparison between pipe and channel flows can be summarized as follows: the mean velocity $U_b{}^+$ in channel flow is between 5% and 12% greater than in pipe flow (the larger the difference, the lower $Re_\tau$), whereas the centerline velocity $U_c{}^+$ is about 4% smaller. The ratio $U_c{}^+/U_b{}^+$ in pipe flow is between 8% and 14% greater than in channel flow (the larger the difference, the lower $Re_\tau$). The skin friction based on the mean velocity in pipe flow is between 8% and 21% larger than in channel flow (the larger the difference, the lower $Re_\tau$), whereas the skin friction based on the centerline velocity is about 10% smaller. The displacement thickness in pipe flow is about 45% larger than in channel flow; the momentum thickness is between 18% and 26% larger (the larger the difference, the higher $Re_\tau$).

**Author Contributions:** Conceptualization, C.D.N. and R.A.; methodology, C.D.N. and R.A.; formal analysis, C.D.N. and R.A.; resources, C.D.N. and R.A.; data curation, C.D.N. and R.A.; writing— original draft preparation, C.D.N.; writing—review and editing, C.D.N. and R.A.; funding acquisition, C.D.N. All authors have read and agreed to the published version of the manuscript.

**Funding:** This research received no external funding.

**Data Availability Statement:** Not applicable.

**Acknowledgments:** We are very grateful to all Colleagues who made their DNS data available to the community.

**Conflicts of Interest:** The authors declare no conflict of interest.

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
