# Peer review of "Comparison of Mean Properties of Turbulent Pipe and Channel Flows at Low-to-Moderate Reynolds Numbers"

_fluids, doi:10.3390/fluids8030097_

Round 1

Reviewer 1 Report

In this manuscript, the authors used the DNS database of turbulent channel flow and pipe flow to fit the wall-normal distributions of the mean velocity with a function called ITM. They then evaluated several quantities determined by the mean velocity distributions of these wall turbulent flows. 

The referee cannot find any scientific significance or deep insight in this manuscript because the authors just summarized differences in the mean property between the two wall turbulence. In addition, the purpose of the present study is unclear. The reviewer, therefore, thinks that the manuscript is not adequate to be published. 

Author Response

To make the purpose clearer, we revised the Abstract and the Section 1, and we have expanded Sections 2 and 3.

We would like to underline that this paper compares fully developed turbulent flows in circular pipes and channels. For channel flow, we deduced the mean velocity profiles using an Indirect Turbulent Model (an approximate analytical model, well tested in a previous paper [30]). For pipe flow we extract the need information from Direct Numerical Simulation database available in open literature. We underline that the comparison is performed at the identical Reynolds numbers (the Indirect Turbulent Model allow us to set any required Reynolds number in the range 110<Retau<2003). In addition, the followed line of reasoning allows us to observe some deviations in the Mean Velocity Profiles extracted from pipe DNS database. These discrepancies, probably due to a dissimilar performance of the numerical schemes, are reflected in global parameters. A brief discussion about these results have been added.

Reviewer 2 Report

The paper is about comparison of mean properties of turbulent pipe and channel flows at low-to-moderate Reynolds numbers. It further explores main authors Indirect Turbulent Model ITM. As in some parts discussion is missing it needs revision. All my detailed 15 comments can be found in the comment windows in attached draft of the paper.

Author Response

Reply point-to-point

Dear Reviewer,

thank you for your comments. Please find our reply:

1 We underline that very important scientific papers have been written in personal way. In this paper we opted for this form.

2 As suggested, other experimental studies have been cited.

3 As suggested, few sentences about Indirect Turbulent Model have been added.

4 As suggested, other reference on the Hellinger distance has been added.

5-9 As suggested, comments about Figs. 2-6 have been added.

10 As suggested, discussion about the obtained results has been added. Furthermore, Table 2 has been expanded.

11-12 As suggested, the source of the differences in the outer layer are stressed.

13 We underline that the ITM data allow us to obtain a comparison at the identical Reynolds number. Previous papers in literature, providing the comparison between turbulent pipe and channel flows via DNS data, consider the comparison at a similar (not identical) Reynolds number. Comparison at identical Reynolds numbers is one of the main features of our paper. According to adopted line of reasoning, in our opinion comparison between turbulent pipe and channel flows via DNS data is out of the paper scope.

14-15 A qualitative discussion about the pipe results has been added. Some quantitative differences are summarized in Section 4.

Reviewer 3 Report

A brief report on the Comparison of mean properties of turbulent pipe and channel flows at low-to-moderate Reynolds numbers was performed. The results reported are interesting however, the comparison of quantities should improve. At the moment most of the results are only compared in numbers without any link to the physical aspect. Please relate the differences observed in the channel and pipe flow through physics. This report can be accepted after incorporating these comments.

Author Response

Dear Reviewer,

thank you for your comments. Please find our reply:

As suggested, the physical aspects about the obtained results have been added. To this purpose, we have revised the Abstract and the Section 1, and we have expanded the Section 2 and the Section 3. Supplementary comments have been added to the figures, and Table 2 has been expanded. The discussion of the results has been deepened.

Round 2

Reviewer 1 Report

The referee cannot understand the value of the present study. However, the authors believe the significance and mention it clearly in the revised manuscript. The referee now accepted the publication for readers who can evaluate the value and are interested in the study. 

Reviewer 2 Report

It looks all my oints are reflected in the revised version of the paper. In my opinion now it can be considered for publication.